# Lipid discovery enabled by sequence statistics and machine learning

Priya M Christensen[1†], Jonathan Martin[1†], Aparna Uppuluri[1], Luke R Joyce[2], Yahan Wei[3], Ziqiang Guan[4]*, Faruck Morcos[1,5,6]*, Kelli L Palmer[1]*

[1]Department of Biological Sciences, University of Texas at Dallas, Richardson, United States; [2]Department of Immunology and Microbiology, University of Colorado Anschutz Medical Campus, Aurora, United States; [3]School of Podiatric Medicine, University of Texas Rio Grande Valley, Harlingen, United States; [4]Department of Biochemistry, Duke University Medical Center, Durham, United States; [5]Department of Bioengineering, University of Texas at Dallas, Richardson, United States; [6]Center for Systems Biology, University of Texas at Dallas, Richardson, United States

*For correspondence:
ziqiang.guan@duke.edu (ZG);
faruckm@utdallas.edu (FM);
kelli.palmer@utdallas.edu (KLP)

[†]These authors contributed equally to this work

Competing interest: The authors declare that no competing interests exist.

## eLife Assessment

This study reports **important** findings on identifying sequence motifs that predict substrate specificity in a class of lipid synthesis enzymes. It sheds light on a mechanism used by bacteria to modify the lipids in their membrane to develop antibiotic resistance. The evidence is **compelling**, with a careful application of machine learning methods, validated by mass spectrometry-based lipid analysis experiments. This interdisciplinary study will be of interest to computational biologists and to the community working on lipids and on enzymes involved in lipid synthesis or modification.

**Abstract** Bacterial membranes are complex and dynamic, arising from an array of evolutionary pressures. One enzyme that alters membrane compositions through covalent lipid modification is MprF. We recently identified that *Streptococcus agalactiae* MprF synthesizes lysyl-phosphatidylglycerol (Lys-PG) from anionic PG, and a novel cationic lipid, lysyl-glucosyl-diacylglycerol (Lys-Glc-DAG), from neutral glycolipid Glc-DAG. This unexpected result prompted us to investigate whether Lys-Glc-DAG occurs in other MprF-containing bacteria, and whether other novel MprF products exist. Here, we studied protein sequence features determining MprF substrate specificity. First, pairwise analyses identified several streptococcal MprFs synthesizing Lys-Glc-DAG. Second, a restricted Boltzmann machine-guided approach led us to discover an entirely new substrate for MprF in *Enterococcus*, diglucosyl-diacylglycerol (Glc$_2$-DAG), and an expanded set of organisms that modify glycolipid substrates using MprF. Overall, we combined the wealth of available sequence data with machine learning to model evolutionary constraints on MprF sequences across the bacterial domain, thereby identifying a novel cationic lipid.

## Introduction

Lipids are ubiquitous and diverse group of amphiphathic compounds that are critical for fundamental biological processes, including the formation of cell membranes, protection, and cargo delivery; energy storage; and cell signaling pathways (*Fahy et al., 2011*). Chemical modifications to lipids, including changes in fatty acid tails (saturated and unsaturated) (*Wang et al., 2018*), modifications of head groups (*Roy et al., 2009*), and alterations to lipid charge (*Roy and Ibba, 2008*), impact these interactions, as well as the physicochemical properties of membranes. Commonly, bacteria synthesize lipids that are negatively charged or neutral (*Sohlenkamp and Geiger, 2016*). Some bacteria modify

**Figure 1.** Chemical structures of lysine lipids. Lys-PG, lysyl-phosphatidylglycerol; Lys-Glc-DAG, lysyl-glucosyl-diacylglycerol; Lys-Glc$_2$-DAG, lysyl-diglucosyl-diacylglycerol.

the negatively charged lipids with amino acids to make them positively charged to confer resistance to cationic antimicrobial peptides and cationic antibiotics (*Weidenmaier et al., 2005*; *Joyce and Doran, 2023*). The bacterial domain is a rich source of natural lipids that are likely coevolved for specific interactions with host tissues and other cells, yet the full extent of this chemical diversity remains unexplored. Finding bacteria that possess unique lipids can be useful for biotechnology. For example, natural lipids found in bacteria can be used for drug delivery. A study by *Gujrati et al., 2014*, used low cytotoxic outer membrane vesicles that contained a modified lipopolysaccharide to deliver siRNA to targeted cancer cells. This delivery mechanism had success and demonstrates the possibility of using naturally occurring bacterial lipids for future drug delivery processes to eukaryotic cells.

One enzyme responsible for modifying lipids in bacteria is the multiple peptide resistance factor (MprF). MprF modifies lipids through the transfer of amino acids from charged tRNAs to the head group of the anionic membrane phospholipid, phosphatidylglycerol (PG) (*Roy and Ibba, 2008*). The MprF protein is comprised of two domains: an N-terminal flippase domain responsible for moving aminoacylated lipids from the inner membrane leaflet to the outer leaflet and a C-terminal synthase domain responsible for the aminoacyl transfer from tRNA to lipids (*Ernst et al., 2009*; *Staubitz et al., 2004*). Recently, in *Streptococcus agalactiae* (Group B *Streptococcus*, GBS) it was found that MprF can modify two different substrates with lysine (Lys) – the glycolipid glucosyl-diacylglycerol (Glc-DAG) and the phospholipid, PG (*Joyce et al., 2022*), generating Lys-Glc-DAG, as well as Lys-PG (*Joyce et al., 2021*; *Joyce et al., 2022*; *Figure 1*). An *S. agalactiae mprF* deletion mutant no longer synthesizes Lys-Glc-DAG or Lys-PG and expression of *S. agalactiae mprF* in the heterologous host *Streptococcus mitis* conferred Lys-Glc-DAG and Lys-PG synthesis to *S. mitis* (*Joyce et al., 2022*). This was the first time MprF was demonstrated to add lysine onto a neutral glycolipid (Glc-DAG).

The synthesis of Lys-Glc-DAG by *S. agalactiae* MprF illustrated a unique lipid substrate and product by the enzyme that had not been characterized before, highlighting the possibility for further exploration into unknown lipids and lipid substrates by MprF of different bacterial species. We sought to investigate the molecular determinants of enzyme specificity and identify other bacterial species that may have this novel specificity. This investigation led us from the standard methodology of BLASTp queries (*Camacho et al., 2009*), which are based on single residue amino acid substitution frequencies, to a more modern statistical method called a restricted Boltzmann machine (RBM) (*Tubiana et al., 2019b*), which captures global statistical patterns within natural sequence data. In this work, we develop a general strategy for combining this high-powered statistical model with lipidomic analysis to discover MprF variants with unique enzymatic activity. This process led us to the discovery of an uncharacterized lipid substrate for MprF, diglucosyl-diacylglycerol (Glc$_2$-DAG), which is present in several strains identified by the parameters of the statistical model. These models also provide interpretable and actionable information for use in further enzyme characterization, and we show that this application is a useful companion tool for experimental work.

**Table 1.** Summary of different MprF variants expressed in *S. mitis* and the lysine lipids they produce. Percentage of amino acid identity and similarity compared to *S. agalactiae* COH1 MprF; data obtained from BLASTp. The lipids each strain synthesize are denoted by a checkmark or an x. Lys-PG, lysyl-phosphatidylglycerol; Lys-Glc-DAG, lysyl-glycosyl-diacylglycerol.

| Bacteria strains | % Amino acid identity (similarity) | Lys-PG | Lys-Glc-DAG |
|---|---|---|---|
| SM61(pGBSMprF) | 100.0 (100.0) | ✓ | ✓ |
| SM61(pFerus) | 61.9 (79.0) | X | ✓ |
| SM61(pSobrinus) | 61.4 (79.0) | X | ✓ |
| SM61(pDownei) | 61.0 (79.0) | X | ✓ |
| SM61(pSalivarius) | 43.1 (66.0) | ✓ | X |
| SM61(pABG5) | - (-) | X | X |

## Results

### Simple pairwise sequence statistics identify streptococcal MprF enzymes that synthesize Lys-Glc-DAG

Utilizing a method based on the amino acid sequence of *S. agalactiae* MprF, we identified four streptococcal MprFs with high sequence identity to *S. agalactiae* COH1 MprF (WP_000733236.1). A BLASTp analysis found that *Streptococcus sobrinus* ATCC 27352 MprF (WP_019790557.1) had 61.4% amino acid identity to the *S. agalactiae* MprF, *Streptococcus salivarius* ATCC 7073 MprF (WP_002888893.1) had 43.09% amino acid identity, *Streptococcus ferus* MprF (WP_018030543.1) had 61.88% amino acid identity, and *Streptococcus downei* (WP_002997695.1) had 61.03% amino acid identity (*Table 1*). Plasmids were designed to express the *S. sobrinus mprF* (pSobrinus), *S. salivarius mprF* (pSalivarius),

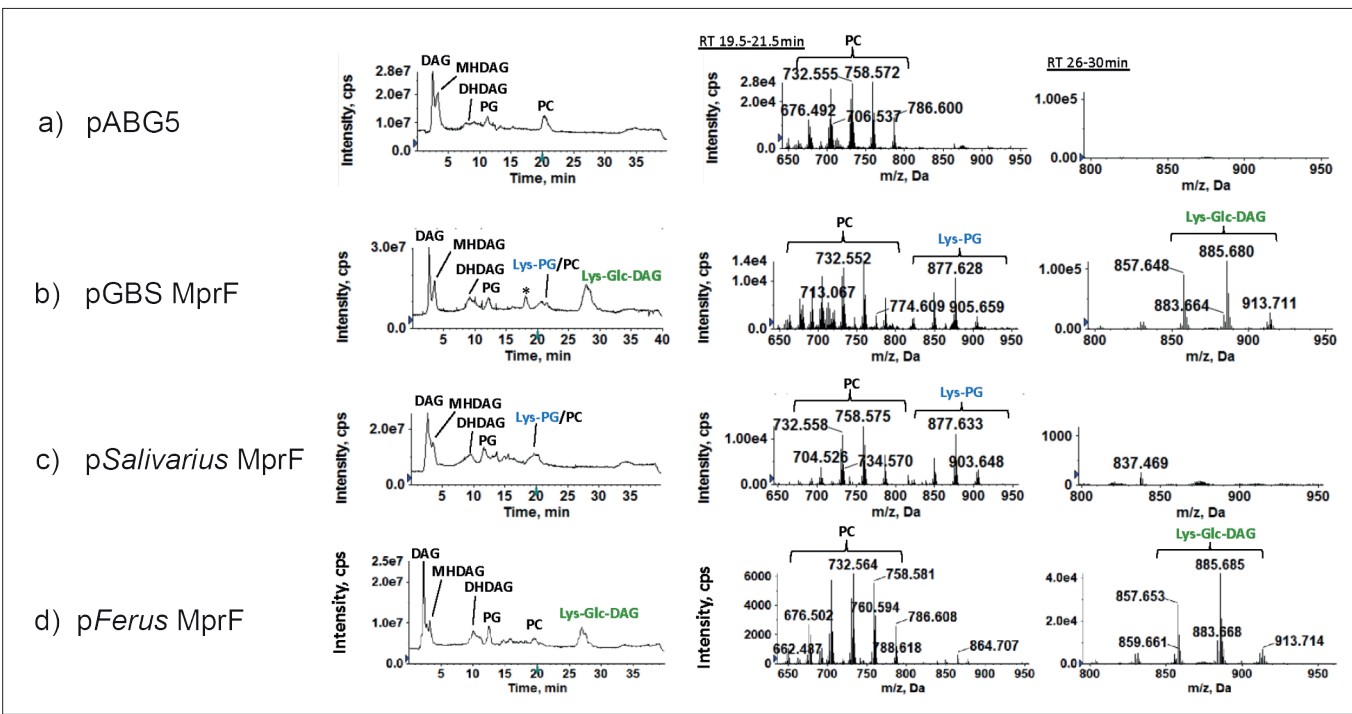

**Figure 2.** Synthesis of lysine lipids (Lys-PG and Lys-Glc-DAG) in *S. mitis* expressing *mprFs* from *S. agalactiae*, *S. salivarus*, and *S. ferus*. (**a**) *S. mitis* NCTC12261 with empty vector control (pABG5) lacks lysine lipids; (**b**) *S. agalactiae mprF* (pGBSMprF) produces both Lys-PG and Lys-Glc-DAG; (**c**) *S. salivarius mprF* produces only Lys-PG; (**d**) *S. ferus mprF* produces only Lys-Glc-DAG. Left panels: total ion chromatograms (TIC); middle panels: mass spectra of retention time 19.5–21.5 min showing Lys-PG and PC; right panels: mass spectra of retention time 26–30 min showing Lys-Glc-DAG. Note: '*' is an extraction artifact due to chloroform used. DAG, diacylglycerol; MHDAG, monohexosyldiacylglycerol; DHDAG, dihexosyldiacylglycerol; PG, phosphatidylglycerol; Lys-PG, lysyl-phosphatidylglycerol; Lys-Glc-DAG, lysyl-glucosyl-diacylglycerol; PC, phosphatidylcholine.

*S. ferus mprF* (pFerus), and *S. downei mprF* (pDownei) genes. The plasmids were transformed into the expression host *S. mitis* NCTC12261 (SM61). The empty vector (pABG5Δ*phoZ*, referred to as pABG5 here) was also transformed (*Figure 2a*). *S. mitis* does not natively encode *mprF* but synthesizes lipids (Glc-DAG; PG) that are substrates for *S. agalactiae* MprF (*Adams et al., 2017*; *Joyce et al., 2019*; *Joyce et al., 2022*), making it an appropriate heterologous host for expressing streptococcal MprFs. Previously, a plasmid expressing *S. agalactiae mprF* (pGBSMprF) (*Joyce et al., 2022*) was generated and transformed into *S. mitis*. Lys-Glc-DAG and Lys-PG presence in SM61 was only possible with the expression of pGBSMprF (*Figure 2b*; *Joyce et al., 2022*).

*S. mitis* lipids of strains expressing the plasmids pSobrinus (*Table 1*), pSalivarius (*Figure 2c*), pFerus (*Figure 2d*), and pDownei (*Table 1*) were analyzed. Lipidomic analysis found that pSobrinus, pDownei, and pFerus conferred synthesis of Lys-Glc-DAG, but not Lys-PG in *S. mitis*. Notably, pFerus confers synthesis of Lys-Glc-DAG at a similar level to *S. agalactiae* MprF (*Figure 2b and d*). In contrast, pSalivarius conferred synthesis of Lys-PG only and not Lys-Glc-DAG. Importantly, *S. salivarius* MprF also synthesized Lys-PG at a level most similar to *S. agalactiae* MprF (*Figure 2b and c*). Lysine-modified lipids synthesized by *S. mitis* strains expressing streptococcal MprFs used in this study are summarized in *Table 1*. In general, streptococcal MprF enzymes with higher amino acid percent (>60%) to *S. agalactiae* MprF conferred Lys-Glc-DAG synthesis to *S. mitis*, but only *S. agalactiae* MprF conferred both Lys-Glc-DAG and Lys-PG synthesis in *S. mitis*.

## RBM can provide sensitive, rational guidance for sequence classification

We set out to understand which specific MprF amino acid residues are involved in enzyme specificity for either the PG or Glc-DAG substrate. To this end, we employed an RBM, a probabilistic graphical model which aims to model the probability of data in a dataset through statistical connections between the features of the data and a set of hidden units (*Smolensky, 1986*).

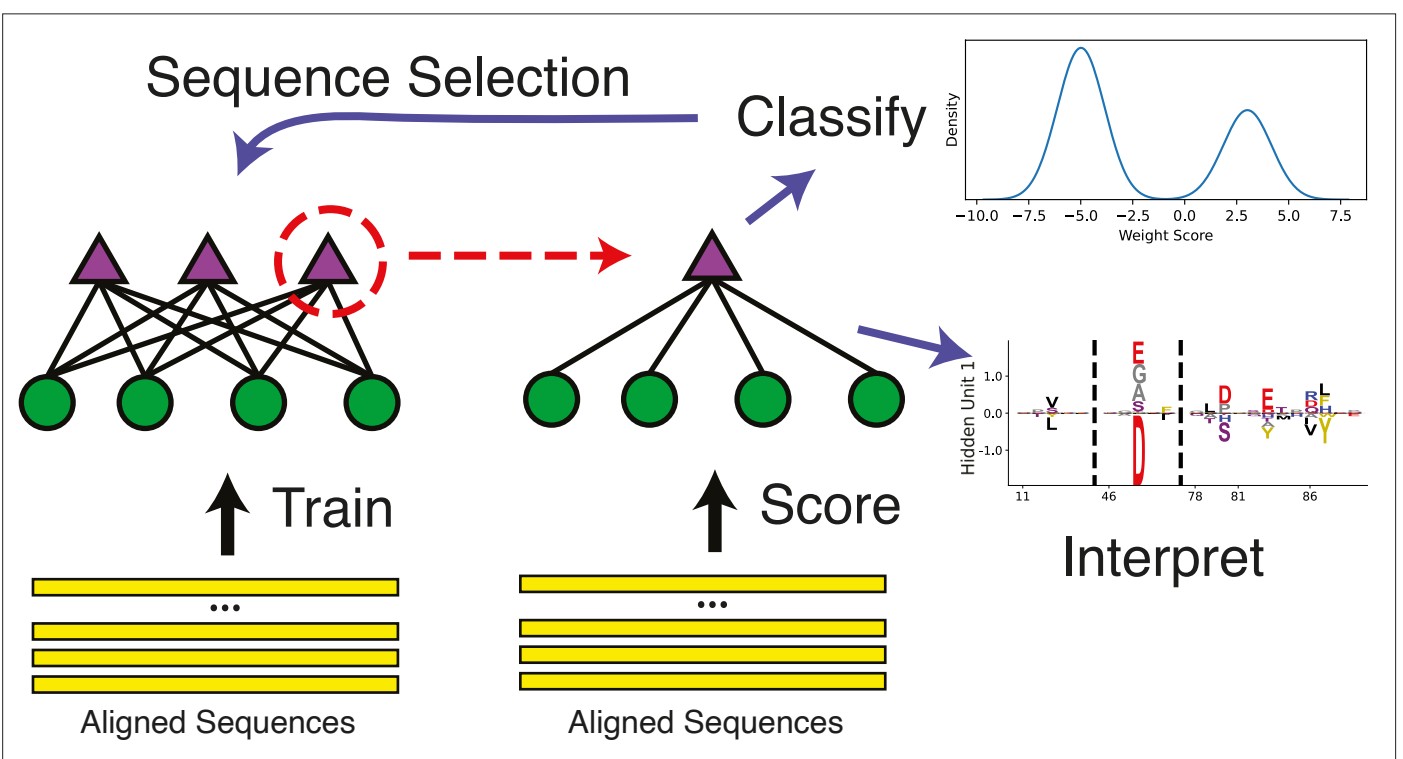

**Figure 3.** Schematic of the restricted Boltzmann machine (RBM) methodology. An aligned set of protein sequences is first used to learn a hidden unit representation that best describes the statistics of the sequence dataset given restrictions on the hidden unit representation. Then, the individual hidden units can be studied to find particular units which allow useful enzyme classification, and additionally, these weights can be meaningfully interpreted as statistically covarying sequence configurations. Additionally, the classification can be used to create filtered datasets to train more models.

This class of models is closely related to another model termed the Boltzmann machine. The Boltzmann machine formulation is closely related to the Potts model from physics, which was successfully applied in biology to elucidate important residues in protein structure and function (***Morcos et al., 2011***), one example being the careful tuning of enzyme specificity in bacterial two-component regulatory systems (***Cheng et al., 2014***; ***Jiang et al., 2021***). The Boltzmann machine formulation from ***Morcos et al., 2011***, termed direct-coupling analysis, stores patterns in the form of a pairwise coupling matrix and local field matrix (shown in ***Equation 2***), which has been useful in understanding how likely pairs of residues are to interact in physical space. We hypothesize that *S. agalactiae* MprF and other MprF enzymes that utilize glycolipid substrates have unique patterns of interacting residues relative to those enzymes that use only PG as a substrate.

While modeling how pairs of residues covary is powerful, many important features of enzyme function like allostery or enzyme specificity (which is the purpose of this analysis) may involve more than two residues (***Goodey and Benkovic, 2008***), and in these situations it can be difficult to connect sets of residues using solely pairwise parameters (***Figliuzzi et al., 2018***). Therefore, a more global approach is required to address our hypothesis. To this end, an RBM has recently been developed for use in the learning of protein sequence data, with special attention given to producing a model which produces sparse, interpretable, and biologically meaningful representations of these higher-order statistical couplings within the hidden units (***Tubiana et al., 2019b***). The design of this RBM can be seen in ***Figure 3***, where the model architecture is represented by purple dots and green triangles. The dots are the 'visible' layer, which take in input sequences and encode them into the 'hidden' layer, where each triangle represents a separate hidden unit. The lines connecting the visible and hidden layers show that each hidden unit can see all the visible units (the statistics are global), but they cannot see any of the other hidden units, meaning the hidden units are mutually independent. This global model with mutually independent hidden units (see the marginal distribution form shown in ***Equation 3***) allows the following useful properties: higher-order couplings between residues (sets of coupled residues instead of pairs); bimodal hidden unit outputs given the training dataset (***Equation 4***); sparse, interpretable configurations of the hidden units ($w_\mu$) (***Equation 5***); and a compositional representation of the hidden unit weights (***Tubiana et al., 2019a***), where input sequences are modeled largely through combinations of hidden units instead of single, highly activating units.

The compositional representation is a critical feature of this method, allowing us to find independently coupled sets of residues which in combination describe these protein sequences, and it is achieved through the particular choices of training parameters. More details on the training parameter choices are found in Materials and methods. In particular, we focus on the scoring of sequences using individual hidden units $w_\mu$:

$$I_\mu(\mathbf{v}) = \sum_i \mathbf{w}_{i\mu}(\mathbf{v}_i)$$

(1)

where $\mathbf{v}$ is an input sequence vectorized through one-hot encoding, to produce a hidden unit activation (a single number) which depends on the input sequence residues at position $i$ and the value of the hidden unit weight matrix for that residue at that position.

We used this methodology to study the sequences in the family of MprF, which is a large protein composed of independent flippase and synthase domains (***Ernst et al., 2009***; ***Slavetinsky et al., 2012***). In our analysis, we restrict ourselves to the cytosolic region (see yellow box in ***Figure 4a***), also known as the aminoacyl-PG synthase domain, which constituted the Pfam family DUF2156 (now renamed LPG_synthase_C) (***Mistry et al., 2021***).

One hidden unit in particular is highlighted, where we find bimodal activation (***Figure 4b***) of the hidden unit (***Figure 4c***) when ***Equation 1*** is evaluated with the training sequences. Interpreting the sequence histograms, a positive value activation in ***Figure 4b*** corresponds to a sequence containing combinations of the residues listed in the positive portion of the weight diagram in ***Figure 4c***, whereas a negative activation corresponds to residues in the negative portion. For display purposes, the residue weights shown in all of our hidden units correspond to the positions where the magnitude of the visible positions $w_{i\mu}$ is greater than some threshold proportional to the largest magnitude position (position 212); the hidden units have values for all possible amino acids in the entire protein length, but they are considerably smaller than this threshold and generally very low.

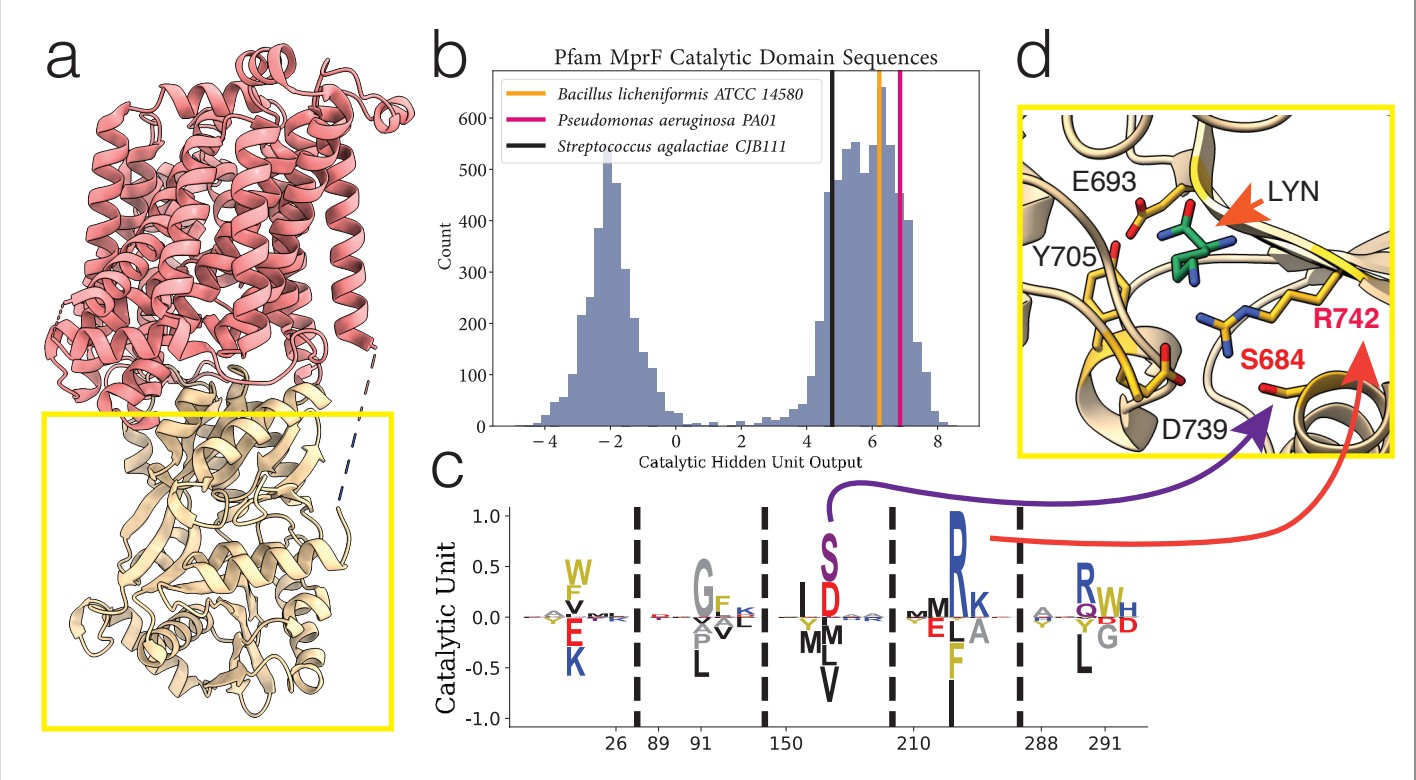

**Figure 4.** Example of hidden unit analysis and usage. (**a**) The structure of PDB:7DUW, with the red colored region being the transmembrane flippase domain and the yellow boxed region the cytosolic domain which we focus on. (**b**) The activations produced by inputting a sequence into a hidden unit, producing a single number as output which corresponds to a summation of negatively and positively weighted residues. Performed on entire training set (histogram in blue), highlighting sequences corresponding to predominantly positive weighted residues. (**c**) Hidden unit from a restricted Boltzmann machine (RBM) trained on the Pfam DUF2156 domain. The MSA positions 152 and 212 correspond to residues S684 and R742, respectively. (**d**) Residues (in yellow) in the MprF cytosolic domain which form the binding pocket for Lys-tRNA$^{Lys}$ (the ligand analogue L-lysine amide shown in green), from PDB:4V36. LYN, L-lysine amide.

The online version of this article includes the following figure supplement(s) for figure 4:

**Figure supplement 1.** The full sequences of our pre-filtered multiple sequence alignment were analyzed to identify which Pfam domains were present in them.

Two positions in particular, 152 and 212, in the positive portion of the hidden unit correspond to residues 684 and 742, which were experimentally demonstrated to be relevant for aminoacylated-PG production by the *Bacillus licheniformis* and *Pseudomonas aeruginosa* cytosolic synthase domain variants (*Hebecker et al., 2015*); in *B. licheniformis* the residues which form a complex with the lysine from $Lys-tRNA^{Lys}$ are shown in *Figure 4c*. We have highlighted these two aforementioned variants as well as the *S. agalactiae* variant for demonstration of sequence scoring through *Equation 1* in *Figure 4a*. We used this hidden unit to create a filtered dataset; only sequences that contained this necessary motif were of interest to us, and we subsequently trained a new model using only sequences with a catalytic hidden unit activation greater than 2 (because it cleanly split the bimodal distribution). We outline this general workflow in *Figure 3*. We also note that the full-length sequences of the sequences included in our new dataset are enriched with sequences which contain the bacterial flippase domain (shown in red in *Figure 4a*), and the sequences we removed rarely contain the flippase domain (see *Figure 4—figure supplement 1* for domain analysis).

## RBM hidden units identify plausible sequence residues for functional characterization

After filtering the dataset and training another model, we set out to find hidden units which could describe MprF specificity. To this end, we identify hidden units which are sparse, have high overall magnitude, and involve residues which could be plausibly linked to the lipid-binding specificity (see

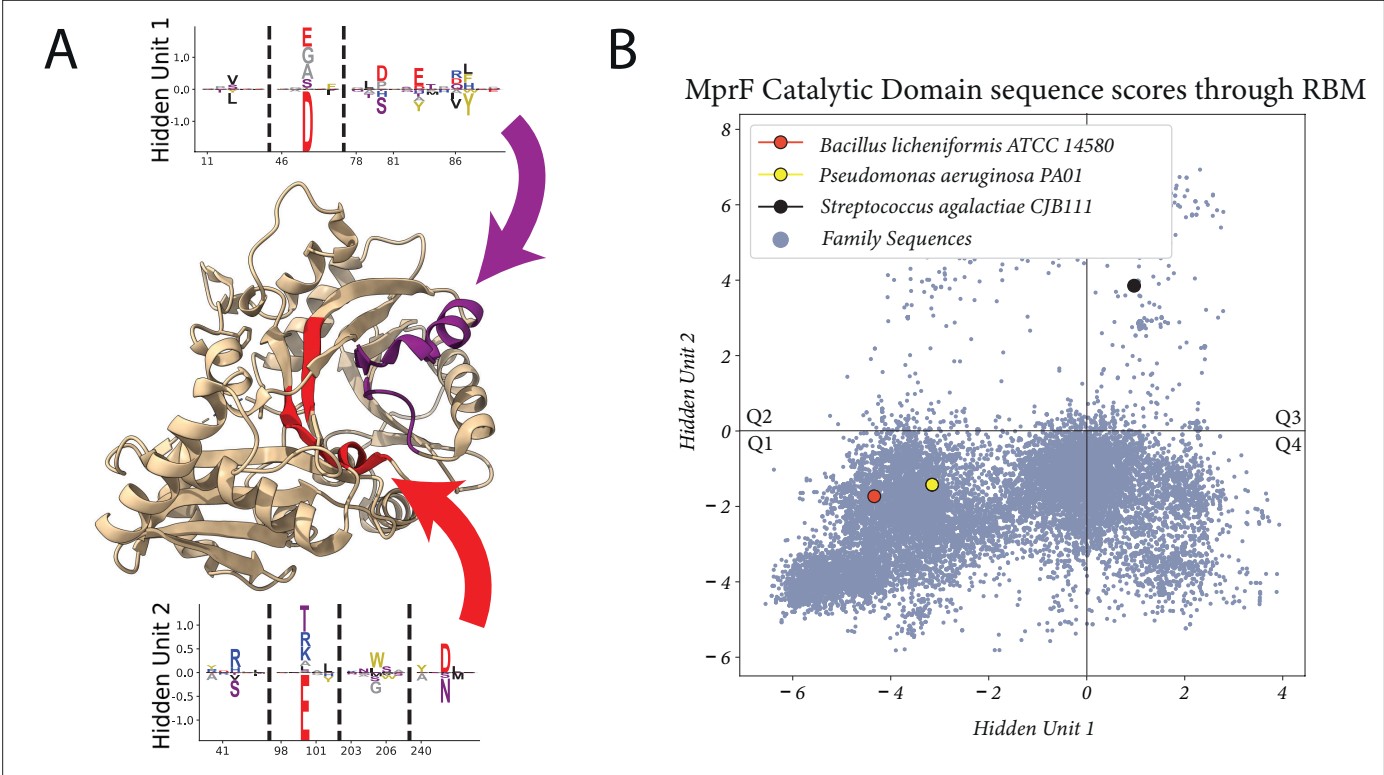

**Figure 5.** Proposed set of hidden units for classifying lipid specificity. (**A**) Two hidden units found in a restricted Boltzmann machine (RBM) trained on the filtered Pfam dataset. The hidden unit residues are highlighted in the PDB:4V34 structure, with arrows pointing to their corresponding residue sets. (**B**) The activations of the hidden units when scoring the sequences used in the training set and sequences from NCBI not used during training (*N*=23,138). *S. agalactiae* produces Lys-Glc-DAG and Lys-PG, while *B. licheniformis* and *P. aeruginosa* produces only aminoacylated-PG. Q1-Q4 are quadrant labels which we refer to throughout the paper. Lys-PG, lysyl-phosphatidylglycerol; Lys-Glc-DAG, lysyl-glucosyl-diacylglycerol.

The online version of this article includes the following figure supplement(s) for figure 5:

**Figure supplement 1.** Assessment of restricted Boltzmann machine (RBM) hidden unit's reliance on total sequence identity in classification.

Materials and methods for more details). Through this search method we identified the two hidden units shown in *Figure 5A*. They both have high sparsity and high magnitude relative to the majority of other hidden units in the model, and importantly they correspond to residues localized to regions of the catalytic domain that had been previously identified through Autodock experiments using a PG molecule with side chains C5:0/C8:0 (*Hebecker et al., 2015*) (highlighted in *Figure 5A*). The scoring of the training set of sequences with these hidden units is shown in *Figure 5B*, and we see a clear separation between the cytosolic domain sequences which produces Lys-Glc-DAG (from *S. agalactiae*) and two domain variants which produce only aminoacylated-PG (from *P. aeruginosa* and *B. licheniformis*). Importantly, these output activations depend on a greatly reduced subset of the full sequence and do not correspond to clustering on total sequence identity (*Figure 5—figure supplement 1*).

Using these two hidden units as sequence classifiers with a potential link to the substrate specificity of cytosolic domain variants, we used *Figure 5B* to propose sequences in quadrant three (Q3), the quadrant occupied by the *S. agalactiae* MprF variant with this novel glycolipid-modifying function. We selected a sequence which was distant in terms of sequence identity from the *Streptococcus* genus (*Supplementary file 1*) and which has been implicated in human pathologies (*Mundy et al., 2000*), *Enterococcus dispar* MprF. We also emphasize that the two hidden units we selected are from a set of 300 total hidden units, and that while other hidden units may differentially classify our three sequences, without the additional link to lipid binding provided by the Autodocking experiment there would be only weak justification for using their activations to make predictions.

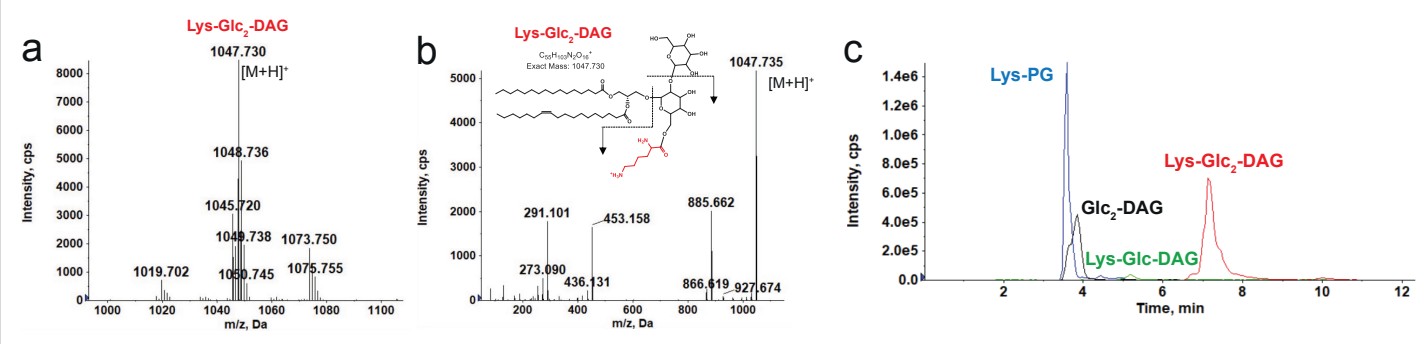

**Figure 6.** Identification of Lys-Glc$_2$-DAG in *E. dispar*. (**a**) Positive ion mass spectrum of Lys-Glc$_2$-DAG species in *E. dispar*. Major Lys-Glc$_2$-DAG species with carbon atoms (before colon) and double bonds (after colon). (**b**) MS/MS product ions and fragmentation scheme of the most abundant Lys-Glc$_2$-DAG ion at *m/z* 1047.73. (**c**) Extracted ion chromatograms of LC/MS of lysine lipids (Lys-PG, Lys-Glc-DAG, Lys-Glc$_2$-DAG) and Glc$_2$-DAG separated on an amino HPLC column. Glc$_2$-DAG, diglucosyl-diacylglycerol; Lys-PG, lysyl-phosphatidylglycerol; Lys-Glc-DAG, lysyl-glucosyl-diacylgylcerol; Lys-Glc$_2$-DAG, lysyl-diglucosyl-diacylglycerol.

The online version of this article includes the following figure supplement(s) for figure 6:

**Figure supplement 1.** Identification of Lys-Glc$_2$-DAG which is highly retentive on a silica HPLC column.

## Enterococcal MprF enzymes synthesize Lys-Glc-DAG, Lys-PG, and a novel cationic glycolipid Lys-Glc$_2$-DAG

*E. dispar* ATCC 51266 was identified as a possible candidate for novel lipid synthesis through the analyses described above. Remarkably, our lipidomic analysis found that *E. dispar* synthesizes a novel cationic lipid, lysyl-diglucosyl(Glc$_2$)-diacylglycerol (Lys-Glc$_2$-DAG) (*Figure 1*, *Figure 6a and b*), as well as Lys-Glc-DAG and Lys-PG (*Figure 6c*, *Figure 6—figure supplement 1*). The structural identification of Lys-Glc-$_2$-DAG was supported by exact mass measurement ([M+H]+ observed at *m/z* 1047.730; calculated *m/z* 1047.731) and tandem MS (*Figure 6b*). Lys-Glc-$_2$-DAG is unusually polar and charged for a lipid; chromatographically, Lys-Glc$_2$-DAG is highly retentive on a silica-based normal phase column and elutes off the column at the very end of the gradient (*Figure 6—figure supplement 1*). The use of an amino-based HPLC column led to much improved elution profiles for Lys-Glc$_2$-DAG and other glyco- and lysine lipids (*Figure 6*).

The discovery of Lys-Glc$_2$-DAG in *E. dispar* led us to analyze other *Enterococcus* strains: *Enterococcus faecium* 1,231,410, *Enterococcus faecalis* T11 and OG1RF *Enterococcus gallinarium* EG2, *Enterococcus casseliflavus* EC10, and *Enterococcus raffinosus* Er676. All *Enterococcus* strains examined were found to synthesize Lys-PG, Lys-Glc-DAG, and Lys-Glc$_2$-DAG at varying levels (*Figure 7a* and *Table 2*). MprF-dependent Lys-PG production by *E. faecalis* and *E. faecium* was previously reported (*Bao et al., 2012*; *Roy and Ibba, 2009*), but cationic glycolipids production has not been previously reported. *E. gallinarium* EG2 and *E. casseliflavus* EC10 encode a single copy of *mprF* gene, while the other enterococci we tested encode two distinct *mprF* genes (*mprF*1 and *mprF*2) (*Bao et al., 2012*; *Roy and Ibba, 2009*). When these MprF sequence variants are plotted using our chosen RBM hidden units, we see that the two single copy strains plot in quadrant two (Q2), while for the two copy sequences, each MprF variant maps to separate coordinates (*Figure 7—figure supplement 1*). Both *E. raffinosus* Er676 MprF variants plot to quadrant two (with the lowest Weight 2 value being 0.34). The *E. faecium* and *E. faecalis* variants have copies plotting directly in quadrant one and variants plotting farther out from the sequence cluster; we note that the two displaced variants are both lacking the characteristic aspartic acid and glutamic acid amino acids at positions 100 and 48 in the hidden units (*Figure 5A*), which are the dominant contributions for determining quadrant one occupancy. To clarify which variants were determinants of the novel cationic lipid production, lipidomic analysis on an *E. faecalis* OG1RF mini-mariner (EfaMarTn) transposon mutant (*Dale et al., 2018*) with a transposon insertion within OG1RF_10760 *mprF* (referred to as *mprF*2 in previous studies [*Bao et al., 2012*]; OG1RF_10760::Tn) revealed that Lys-PG, Lys-Glc-DAG, and the newly identified Lys-Glc$_2$-DAG were absent when the synthase domain of the gene was disrupted (*Figure 7b*, *Supplementary file 1*). This led us to the conclusion that *E. faecalis mprF*2 is necessary for the synthesis of the three Lys-lipids in *E. faecalis*. When the empty vector (pABG5) was transformed into OG1RF_10760::Tn, no Lys-lipids

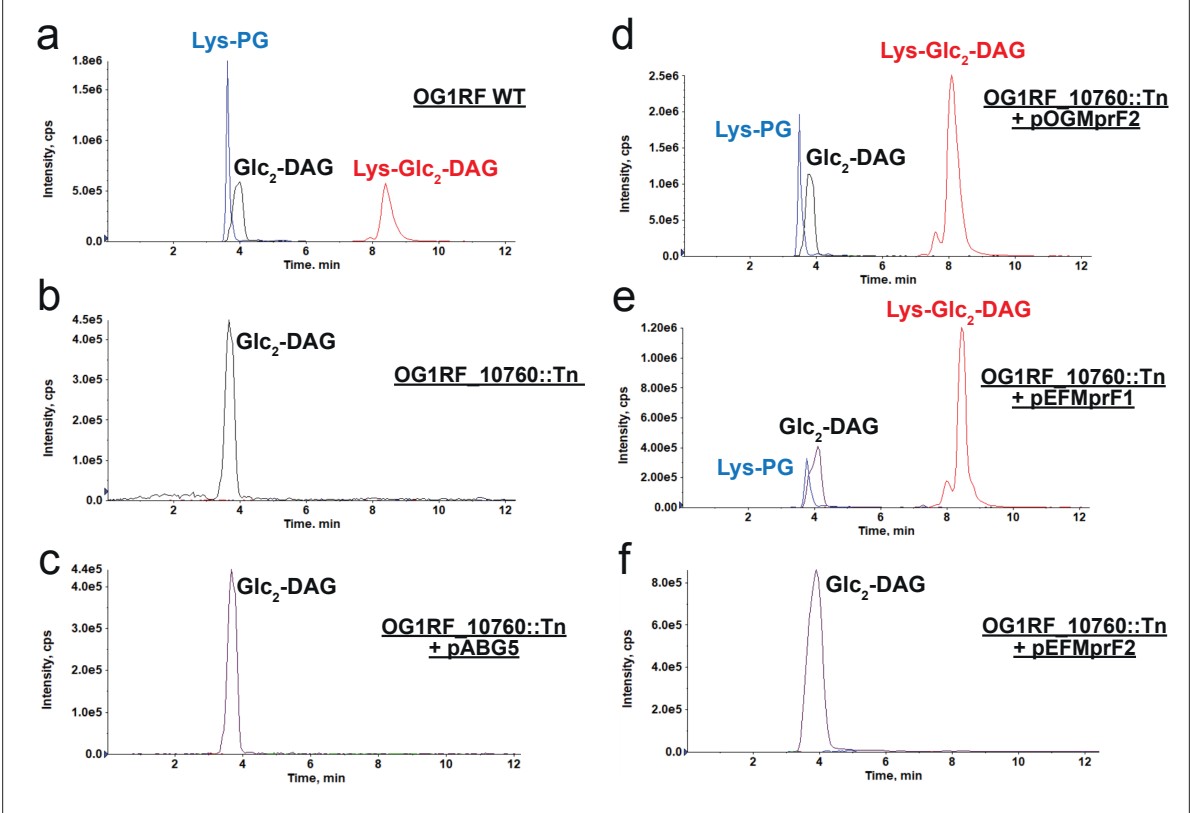

**Figure 7.** *E. faecalis* MprF2 and *E. faecium* MprF1 confer Lys-Glc$_2$-DAG synthesis. (**a**) OG1RF-WT; (**b**) OG1RF_10760::Tn lacks lysine lipids; (**c**) OG1RF_10760::Tn+pABG5 lacks lysine lipids; (**d**) OG1RF_10760::Tn+pOGMprF2 restores lysine lipids; (**e**) OG1RF_10760::Tn+pEFMprF1 restores lysine lipids; (**f**) OG1RF_10760::Tn+pEFMprF2 lacks lysine lipids. Expression of OGMprF2 and EFMprF1 in OG1RF_10760 Tn mutant restores Lys-Glc$_2$-DAG synthesis. Shown are the extracted ion chromatograms of lysine lipids (Lys-PG, Lys-Glc$_2$-DAG) and Glc$_2$-DAG separated on an amino HPLC column. Note: Lys-Glc-DAG was found in trace amounts or missing from lipid extractions. Glc$_2$-DAG, diglucosyl-diacylglycerol; Lys-PG, lysyl-phosphatidylglycerol; Lys-Glc$_2$-DAG, lysyl-diglucosyl-diacylglycerol.

The online version of this article includes the following figure supplement(s) for figure 7:

**Figure supplement 1.** All *Enterococcus* sequences analyzed in the course of this study.

synthesis was observed as expected (*Figure 7c*). This phenotype was complemented by expression of *E. faecalis mprF2* from a plasmid (pOGMprF2) (*Figure 7d*). Additionally, expression of *E. faecium mprF1* (EFTG_00601) (pEfMprF1) restored Lys-PG and Lys-Glc$_2$-DAG synthesis to the *E. faecalis* OG1RF_10760::Tn strain (*Figure 7e*), while *E. faecium mprF2* (EFTG_02430) did not rescue Lys-lipid synthesis (*Figure 7f*). Taken together, these data indicate that *E. faecalis* MprF2 (OG1RF_10760) and *E. faecium* MprF1 (EFTG_00601), like *S. agalactiae* MprF, act on both phospholipid and glycolipids substrates, additionally synthesizing the novel cationic glycolipid, Lys-Glc$_2$-DAG.

## Hidden unit 2 is correlated with Glc$_N$-DAG substrate activity in MprF

When we started our RBM analysis, we had substrate specificity information for only the quadrants one and three, and through identifying the novel substrate in *E. dispar* we expanded our testing to quadrant two by testing *Enterococcus* strains. To expand our RBM analysis, we tested specific sequences which were found in quadrant four (Q4). We conducted lipid analysis on *Ligilactobacillus salivarius* ATCC 11741, *Levilactobacillus brevis* ATCC 14869, *Lacticaseibacillus rhamnosus* ATCC 7469, *Lactobacillus casei* ATCC 393, and *Lacticaseibacillus paracasei* ATCC 25302. All strains tested synthesize Lys-PG (*Table 2*). Only *L. brevis* and *L. paracasei* synthesized Lys-Glc$_2$-DAG, *L. paracasei* synthesized low levels of Lys-Glc$_2$-DAG. An additional quadrant one (Q1) strain *Exiguobacterium acetylicum* UTDF19-27C was analyzed by lipidomics. It was found to synthesize only Lys-PG. This MprF falls near *Staphylococcus aureus* RN4220 and *Bacillus subtilis* 168 in our plot, both of which synthesize only

**Table 2.** Table of all strains studied, the quadrant they occupy, and the lipids they synthesize.
** trace amounts Lys-Glc-DAG present in lipid extractions. * indicates heterologous expression of *mprF* in *S. mitis*. Lys-Glc-DAG, lysyl-glucosyl-diacylglycerol; Lys-Glc$_2$-DAG, lysyl-diglucosyl-diacylglycerol; Lys-PG, lysyl-phosphatidylglycerol.

| | Bacterial strains | Lys-Glc-DAG | Lys-Glc$_2$-DAG | Lys-PG |
|---|---|---|---|---|
| | *Bacillus subtilis* 168 | X | X | ✓ |
| | *Bacillus licheniformis* ATCC 14580 | X | X | ✓ |
| | *Staphylococcus aureus* RN4220 | X | X | ✓ |
| | *Exiguobacterium acetylicum* UTDF19-27C | X | X | ✓ |
| | *Enterococcus faecalis* T11/OG1RF** | ✓ | ✓ | ✓ |
| Q1 | *Enterococcus faecium* 1,231,410 | ✓ | ✓ | ✓ |
| | *Enterococcus raffinosus* Er676 | ✓ | ✓ | ✓ |
| | *Enterococcus gallinarum* EG2 | ✓ | ✓ | ✓ |
| Q2 | *Enterococcus casseliflavus* EC10 | ✓ | ✓ | ✓ |
| | *Streptococcus salivarius** ATCC 7073 | X | X | ✓ |
| | *Enterococcus dispar* ATCC 51266 | ✓ | ✓ | ✓ |
| | *Streptococcus agalactiae* CJB111/COH1 | ✓ | X | ✓ |
| | *Streptococcus sobrinus** ATCC 27352 | ✓ | X | X |
| | *Streptococcus downei** (WP_002997695.1) | ✓ | X | X |
| Q3 | *Streptococcus ferus** (WP_018030543.1) | ✓ | X | X |
| | *Ligilactobacillus salivarius* ATCC 11741 | X | X | ✓ |
| | *Lacticaseibacillus rhamnosus* ATCC 7469 | X | X | ✓ |
| | *Lactobacillus casei* ATCC 393 | X | X | ✓ |
| | *Levilactobacillus brevis* ATCC 14869 | X | ✓ | ✓ |
| Q4 | *Lacticaseibacillus paracasei* ATCC 25302 | X | ✓ | ✓ |

Lys-PG (*Peschel et al., 2001*; *Slavetinsky et al., 2012*) (and experimentally confirmed in this study). This further supports that strains located in quadrant one of *Figure 8* are capable of Lys-PG synthesis.

Summaries of all functional species/variants tested and the Lys-lipids they produced are shown in *Table 2*, and plotted in the two RBM hidden units in *Figure 8*. We list for each MprF variant which of the three tested lipids it acts on, and in *Figure 8* we indicate whether a variant has only PG as a substrate or if it operates on Glc$_N$-DAG as a substrate irrespective of its PG activity. With this plot, we can see that sequences which plot with positive activations in hidden unit 2 are more likely to have Glc$_N$-DAG as a substrate (8/9), and sequences with negative values predominantly use PG (7/11), with a Fisher's exact test p-value of 0.028 (*Supplementary file 2*). Hidden unit 1, however, is not correlated with Glc$_N$-DAG specificity, with a Fisher's exact p-value of 0.67. Therefore, we conclude that hidden unit 2, and thus the high magnitude sequence positions identified by it, are correlated with Glc$_N$-DAG specificity.

## Discussion

The utilization of the *S. agalactiae* MprF amino acid sequence as a query for BLASTp was a simple method to identify various streptococcal MprFs and further our understanding of MprF specificity. This allowed us to identify three MprFs that can synthesize the highly cationic Lys-Glc-DAG. Importantly, although highly similar to *S. agalactiae* MprF, this method did not identify other streptococcal MprFs capable of lysine addition to both glycolipid and phospholipid substrates (generating Lys-Glc-DAG and Lys-PG) nor did it lead to the identification of a novel cationic glycolipid found in *Enterococcus*. Sequence logos (*Schneider and Stephens, 1990*) may identify important sequence features, but largely in instances where amino acids are highly conserved. When amino acids must interact to

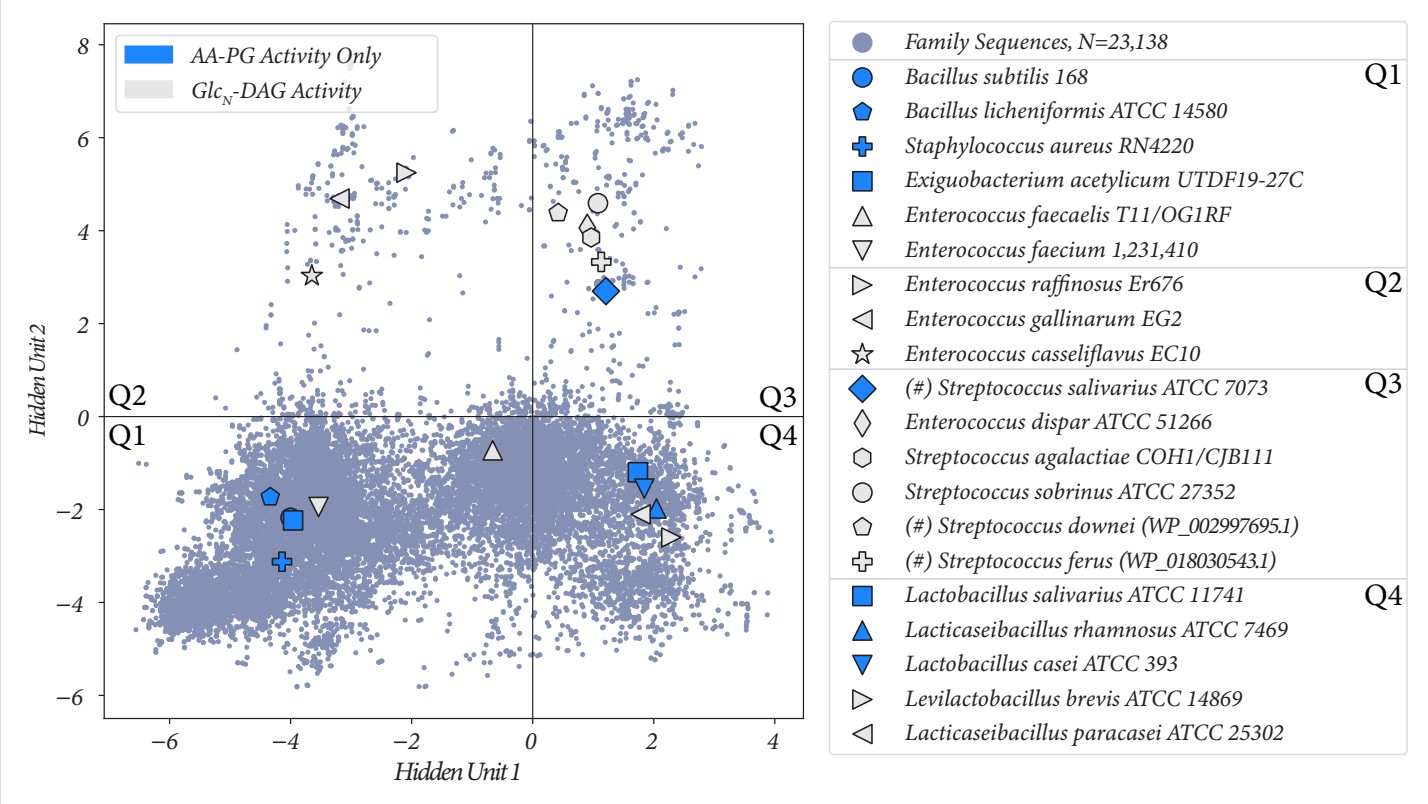

**Figure 8.** The two proposed hidden unit activations with all of the sequences from **Table 2** labeled. Protein ID of highlighted sequences are listed in **Supplementary file 1**. (#) indicates the lipid activity was confirmed through heterologous expression.

produce a specific function, yet the exact identity of those amino acids is not a requirement, methods which model the strength of amino acid coevolution in sequence sets can find important signals which in the sequence logo might appear to be negligible. For this reason we used the RBM method (**Tubiana et al., 2019b**), which can model statistical connectivity of residues at multiple sites in the protein sequence.

The RBM can be used in a number of ways to study sequence data. First, it allowed us to filter the sequence data using hidden units (**Figure 4**); the two hidden units which formed the basis of our specificity analysis were not found in models trained on the Pfam dataset when it was initially pulled from Pfam, and this was potentially due to the presence of sequences without relevant catalytic function adding noise to the dataset. The most prominent feature for our purposes is the ability to use the hidden units learned through RBM training to classify sequences. The bimodal nature of this particular model provides a relatively simple interpretation which is straightforward to use for prediction.

We show in **Figure 5** that the combination of clustering and interpretable hidden unit structure allows us to identify a small subset of residues within a structure, grounding our statistical clustering in MprF's structural features. The combination of experimental evidence and evolutionary information provided a strong rationale for selecting organisms for further study, ultimately leading to the discovery of novel cationic lipid biosynthesis in *E. dispar* (**Figure 6**, **Figure 6—figure supplement 1**).

One goal of this study was to find the structural determinants of $Glc_N$-DAG activity by MprF, and we found that hidden unit 2 from our RBM analysis was predictive of this. However, there are notable exceptions such as the *E. faecium* allele EGTF_00601, which has sequence features placing it in quadrant one yet has $Glc_N$-DAG activity. Therefore, identifying the exact sequence determinants of $Glc_N$-DAG specificity remains to be fully elucidated. One explanation could be our exclusion of the N-terminal flippase domain and the non-cytosolic region of the C-terminal domain in the RBM analysis, where important residue interactions determining specificity could occur and is under further investigation. Additionally, a challenge in experimental validation will be to standardize the method of studying *mprF* alleles from diverse organisms. When using native organisms, lipids produced may vary

during the growth phase of the organisms, the media they are grown in, and whether a membrane stressor is present (i.e. inducible production of specific lipids). For this study we used stationary phase cultures of equal volume and bacteria were grown in their respective standard laboratory media. The use of a heterologous host for expression of *mprF* alleles is a method of standardization, but may result in not identifying novel lipids that would be found in their native bacteria since the appropriate lipid substrates may not be present (*Joyce et al., 2019*; *Joyce et al., 2021*). A combination of both approaches (native organisms and heterologous hosts) may be required. Additionally, spike-in controls might be implemented to help us to more quantitatively understand the output specificity of these MprF variants. Ultimately, mutational studies of the residues identified by the RBM will allow for identification of the residues critical for specificity, though likely the residues will depend in part on the primary sequence being altered.

The general method of RBM-guided exploration has applications in enzyme design, for example, in enhancing methods like directed evolution (*Arnold, 2018*). Concretely, the residues identified through hidden units can be thought of as templates for mutational exploration, greatly reducing the search space for assessing which residues are critical for specificity determination. Enzymes like MprF that produce lipids of modified charge have potential applications in biotechnology and therapeutics, where the design of lipid nanoparticles with different charge and chemical properties of the head group have important physiological implications (*Carrasco et al., 2021*). Notably, the identification of Lys-Glc$_2$-DAG opens new research avenues, particularly in the fields of antibiotic resistance and host-pathogen interactions, wherein lipid modification and cationic lipids play key roles. With respect to human health, it would be relevant to investigate whether Lys-Glc$_2$-DAG plays a role in enterococcal resistance to the last-line membrane-active antibiotic, daptomycin.

## Materials and methods

### Bacterial strains and growth conditions

See *Supplementary files 1 and 3* for a full list of bacterial strains used in this study. *S. mitis* NCTC12261 (referred to as SM61 here) (*Kilian et al., 1989*) was grown in Todd Hewitt Broth (THB) at 37°C with 5% CO$_2$. *Escherichia coli* DH5α (*Taylor et al., 1993*) and *B. subtilis* 168 (*Kunst et al., 1997*) were grown at 37°C with shaking at 225 rpm in lysogeny broth. *S. aureus* RN4220 (*Kreiswirth et al., 1983*) was grown at 37°C with shaking at 220 rpm in Trypic Soy Broth. *S. agalactiae* COH1 ATCC BAA-1176 (*Kuypers et al., 1989*) and CJB111 (*Faralla et al., 2014*; *Spencer et al., 2021*) were grown in THB at 37°C. *E. faecium* 1,231,410 (*Palmer et al., 2010*), *E. faecalis* T11 (*McBride et al., 2007*) and OG1RF (*Gold et al., 1975*), *E. gallinarium* EG2 (*Palmer et al., 2010*), *E. casseliflavus* EC10 (*Palmer et al., 2010*), and *E. raffinosus* Er676 (*Sharon et al., 2023*) were grown at 37°C in Brain Heart Infusion (BHI). *E. dispar* ATCC 51266 (*Collins et al., 1991*) was grown at 30°C in BHI. *E. acetylicum* UTDF19-27C was grown in BHI at 37°C. *L. salivarius* ATCC 11741 (*Drucker, 1979*), *L. rhamnosus* ATCC 7469 (*Collins et al., 1989*), *L. casei* ATCC 393 (*Hansen and Lessel, 1971*), and *L. paracasei* ATCC 25302 (*Collins et al., 1989*) were grown in Lactobacilli MRS Broth at 37°C and 5% CO$_2$. *L. brevis* ATCC 14869 (*Rogosa and Hansen, 1971*) was grown in Lactobacilli MRS Broth at 30°C, 5% CO$_2$. *B. licheniformis* ATCC 14580 was grown in nutrient broth at 37°C with shaking at 225 rpm (*Chester, 1901*). A transposon mutant of *E. faecalis* OG1RF from *Dale et al., 2018*, with a mini mariner transposon (EfaMarTn) insertion in OG1RF10760 (OG1RF10760::Tn) was grown on BHI supplemented with chloramphenicol at a concentration of 15 µg/mL. The mutant was confirmed to be erythromycin-sensitive. The transposon mutant location was confirmed through Sanger sequencing performed by the Genome Center at the University of Texas at Dallas (Richardson, TX, USA).

Where appropriate for plasmid selection, kanamycin (Sigma-Aldrich) was added. For *E. coli*, a concentration of 50 µg/mL was used, for *S. mitis* a concentration of 300 µg/mL, and for *E. faecalis* a concentration of 500 µg/mL was used.

### Routine molecular biology procedures

Genomic DNA was extracted as done previously in *Adams et al., 2015*, and *Manson et al., 2003*. All PCRs used Phusion polymerase (Thermo Fisher Scientific) and Phusion 5x HF buffer (Thermo Fisher Scientific) in a Veriti PCR machine (Applied Biosystems). List of primers used are found in *Supplementary file 4*. Gibson assemblies were completed using a 2x HI-FI Assembly master mix following

the manufacturer's protocol (New England Biolabs). PCR clean-up was done using the GeneJET PCR Purification Kit (Thermo Fisher Scientific) per the manufacturer's protocol. Plasmid extractions were performed per manufacturer's protocol using the GeneJET Plasmid miniprep kit (Thermo Fisher Scientific). All plasmid constructs were confirmed by Sanger sequencing at the Massachusetts General Hospital CCIB DNA Core facility or by Illumina sequence at SeqCenter. DNA concentrations were measured using NanoDrop (Thermo Fisher Scientific) or Qubit 2.0 (Invitrogen by Life Technologies). Optical density at 600 nm (OD600 nm) was measured in a disposable cuvette (Thermo Fisher Scientific) using a spectrophotometer (Thermo Scientific Genesys 30).

Gibson assembly was performed using pABG5 as previously described (*Joyce et al., 2022*). Transformation of plasmids into *E. coli*, *S. mitis*, and *E. faecalis* was performed as previously described in ; *Joyce et al., 2022Joyce et al., 2019*; *Joyce et al., 2022*; *Salvadori et al., 2016*; *Shepard and Gilmore, 1995*.

## Acidic Bligh-Dyer lipid extractions

Bacteria were grown for approximately 16 hr overnight in 15 mL of an appropriate culture medium. After growth, $OD_{600\ nm}$ measurements were taken, and cells were pelleted at 4280 × *g* for 5 min at room temperature in a Sorvall RC6+ centrifuge. The supernatant was tipped out, and the cell pellet was washed and resuspended in 1× phosphate buffered saline (PBS). The cells were pelleted again, and all the supernatant was aspirated out. The cell pellet was stored at –80°C until lipid extraction. An acidic Bligh-Dyer lipid extraction was performed as previously reported (*Joyce et al., 2019*; *Joyce et al., 2021*). Briefly, cell pellets were resuspended in 0.8 mL of 1x Dulbecco's PBS (Sigma-Aldrich). Cells were transferred to a 9 mL glass tube with a Teflon-lined cap (Pyrex). 1 mL chloroform (MilliporeSigma) and 2 mL methanol (Thermo Fisher Scientific) were added to create the single-phase Bligh-Dyer. Tubes were vortexed every 5 min for 20 min, then centrifuged at 500 × *g* for 10 min at room temperature. The supernatant was transferred to a new 9 mL glass tube. 100 µL of hydrochloric acid (Thermo Fisher Scientific) was added followed by 1 mL of chloroform and 0.9 mL of 1x Dulbecco's PBS to create the two-phase Bligh-Dyer. Tubes were gently mixed and centrifuged at 500 × *g* for 5 min at room temperature. After centrifuging, the bottom layer was extracted to a new tube and dried under nitrogen gas and stored at –80°C prior to lipidomic analysis. Lipid analyses were repeated in biological triplicate aside, from OG1RF_10760::Tn extractions, which were performed once.

## Analysis of lysine lipids by an amino column-based LC-ESI MS: a new method

An amino column-based normal phase LC-ESI MS (liquid chromatography electrospray ionization-mass spectrometry) was performed using an Agilent 1200 Quaternary LC system coupled to a high-resolution TripleTOF5600 mass spectrometer (Sciex, Framingham, MA, USA). A Unison UK-Amino column (3 µm, 25 cm ×2 mm) (Imtakt USA, Portland, OR, USA) was used. Mobile phase A consisted of chloroform/methanol/aqueous ammonium hydroxide (800:195:5, vol/vol/vol). Mobile phase B consisted of chloroform/methanol/water/aqueous ammonium hydroxide (600:340:50:5, vol/vol/vol/vol). Mobile phase C consisted of chloroform/methanol/water/aqueous ammonium hydroxide (450:450:95:5, vol/vol/vol/vol). The elution program consisted of the following: 100% mobile phase A was held isocratically for 2 min and then linearly increased to 100% mobile phase B over 8 min and held at 100% B for 5 min. The LC gradient was then changed to 100% mobile phase C over 1 min and held at 100% C for 3 min, and finally returned to 100% A over 0.5 min and held at 100% A for 3 min. The total LC flow rate was 300 µL/min. The MS settings were as follows: ion spray voltage (IS)=5000 V, curtain gas (CUR)=20 psi, ion source gas 1 (GS1)=20 psi, de-clustering potential (DP)=50 V, and focusing potential (FP)=150 V. Nitrogen was used as the collision gas for MS/MS experiments. Data acquisition and analysis were performed using Analyst TF1.5 software (Sciex, Framingham, MA, USA).

## Generation of *mprF* expression plasmids

The *S. downei mprF* nucleotide sequence (Locus tag HMPREF9176_RS03810) and *S. ferus mprF* nucleotide sequence (Locus tag A3GY_RS0106165) were used to design synthetic geneblocks (GeneWiz from Azenta Life Sciences). A 20 nucleotide 5' extension and a 20 nucleotide 3' extension complementary to the pABG5 plasmid were added to enable Gibson assembly of the inserts with pABG5. Sequences for the geneblocks can be found in *Supplementary file 5*. The *S. salivarius mprF* (Locus

tag SSAL8618_04345) was amplified from *S. salivarius* ATCC 7073. A 20 nucleotide 5' extension and a 20 nucleotide 3' extension complementary to the pABG5 plasmid was added to the insert. The same protocol was used for generation of pSobrinus, pEFMprF1, pEFMprF2, and pOGMprF2 with slight modifications. *S. sobrinus mprF* (Locus tag DLJ52_05040) was amplified from *S. sobrinus* ATCC 27352. *E. faecium mprF1* (Locus tag EFTG_00601) and *E. faecium mprF2* (Locus tag EFTG_02430) were amplified from *E. faecium* 1, 231, 410. *E. faecalis mprF2* (Locus tag OG1RF_10760) was amplified from *E. faecalis* OG1RF.

## Boltzmann machines and RBM for protein sequences

### Formulation of models

The Boltzmann machine, as described in *Morcos et al., 2011*, is defined as:

$$P(\mathbf{v}) = \frac{1}{Z} \exp \left( \sum_{i<j} e_{ij}(v_i, v_j) + \sum_i h_i(v_i) \right) \tag{2}$$

which is a probability distribution defined on the pairwise interactions (the couplings $e_{ij}$) between the positions ($v_n$) in an input sequence ($\mathbf{v}$) and a local field term $h_i$. The pairwise coupling matrix is akin to a covariance matrix, and the local fields are similar to single site frequency measures.

The joint probability distribution of the RBM as described in *Tubiana et al., 2019b*, is as follows:

$$P(\mathbf{v}, \mathbf{h}) = \frac{1}{Z} \exp \left( \sum_{i=1}^{N} g_i(v_i) - \sum_{\mu=1}^{M} \mathcal{U}_\mu(h_\mu) + \sum_{i,\mu} h_\mu w_{i\mu}(v_i) \right) \tag{3}$$

Here, the vector $\mathbf{v}$ again represents the input sequence data, $g(v_i)$ is a local field which controls the conditional probability of the input data, $\mathcal{U}_\mu$ is the hidden unit potential/activation, and the terms $h_i$ and $w_{i\mu}(v)$ couple the input variables with the hidden unit variables. In this way, interactions between residues are mediated by a relatively smaller set of hidden units which do not interact directly with each other, leading to a bipartite structure as opposed to the Boltzmann machine's fully connected pair-based formulation.

Particularly important is the potential function $\mathcal{U}$, which in this work is the dRELU function defined as:

$$\mathcal{U}_\mu(h) = \frac{1}{2} \gamma_{\mu,+} \, h_+^2 + \frac{1}{2} \gamma_{\mu,-} \, h_-^2 + \theta_{\mu,+} \, h_+ + \theta_{\mu,-} \, h_-$$
$$h_+ = \max(h, 0) \, , \; h_- = \min(h, 0) \tag{4}$$

The four parameters $\gamma_{\mu,\pm}, \theta_{\mu,\pm}$ are learned through the inference procedure, and allow the positive and negative components of the hidden units $w_\mu$ to be separately gated, which allows learning of bimodal distributions for the activations $h_\mu$. Another key feature is the sparsity regularization on the weights applied during model inference:

$$L2/L1 = \frac{\lambda_1^2}{2qN} \sum_\mu \left( \sum_i \left| w_{i\mu}(v_i) \right| \right)^2 \tag{5}$$

where $q$ is the number of amino acids plus a gap character (21) and the hyperparameter $\lambda_1^2$ can be increased to induce sparsity and lower weight magnitude. *Equations 4 and 5* together guide the representation of the hidden units to bimodal outputs with interpretable features.

### Dataset acquisition and preprocessing

Multiple sequence alignment used for model training was the DUF2156 domain acquired from Pfam (*Mistry et al., 2021*) (in later revisions renamed LPG_synthase_C). Starting with this MSA, it was processed to include only amino acid characters and gaps, excluding sequences with ambiguous or non-standard characters. Additionally, sequences with contiguous strings of gaps greater than 20% of the full sequence's length were removed. This left 11,507 sequences with a length of 298. After the data cleaning process described in *Figure 4*, a total of 7865 sequences were used to train our final model.

## Model training

The model previously described (*Tubiana et al., 2019b*) was utilized, specifically the Python 2.7 version freely available on GitHub (https://github.com/jertubiana/ProteinMotifRBM; *Tubiana, 2019*). Extensive testing of the combinations of the number of hidden units, $L2/L1$ regularization, and learning rate were performed. Models were trained in triplicate with different random seeds, as the training procedure converges to slightly different weight values depending on their initial random configuration. Training was considered successful when the model consistently produced similar sets of weights across different random seeds, and the weights had the previously mentioned sparsity and magnitude indicating compositional representation. The final model used in this work was trained for 2000 epochs with a learning rate of 0.1, a learning rate decay of 0.33, $L2/L1$ parameter $\lambda_1^2$ set to 1.0, and 300 hidden units.

## Model weight selection

To find hidden units which were relevant to function we computed the $L1$ norm of each hidden unit individually, then ranked them. The hidden units with the largest magnitude were looked at first, and these were typically the sparsest of all hidden units. Hidden units were chosen which involved sequence coordinates implicated in our function of interest. Specifically, locations identified through Autodock (*Hebecker et al., 2015*) where the lipid was likely to interact, and a small radius around this region to select a small set of coordinates. We chose the only hidden units which had both overlap with multiple residues in this chosen radius and predictive power (separation) for the three examples we had to start with.

## Software accessibility

DUF2156 domain sequences were retrieved from Pfam33 (*Mistry et al., 2021*). Raw sequence datasets are available as Supplementary Material.

# Acknowledgements

JM and FM acknowledge support from the National Institutes of Health (NIH R35GM133631). FM acknowledges support from the National Science Foundation CAREER award (MCB-1943442). We acknowledge support from the National Institutes of Health grant R01AI178692 (ZG, FM, and KP) and R01AI148366 (ZG and KP). KP acknowledges support from the Cecil H and Ida Green Chair in Systems Biology Science. LRJ acknowledges support from the American Heart Association (23POST1013835).

# Additional information

## Funding

| Funder | Grant reference number | Author |
| --- | --- | --- |
| National Institutes of Health | R35GM133631 | Jonathan Martin<br>Faruck Morcos |
| National Science Foundation | MCB-1943442 | Faruck Morcos |
| National Institutes of Health | R01AI178692 | Ziqiang Guan<br>Faruck Morcos<br>Kelli L Palmer |
| Cecil and Ida Green Foundation | | Kelli L Palmer |
| American Heart Association | 10.58275/AHA.23POST1013835.pc.gr.161273 | Luke R Joyce |

The funders had no role in study design, data collection and interpretation, or the decision to submit the work for publication.

## Author contributions
Priya M Christensen, Data curation, Formal analysis, Validation, Investigation, Visualization, Writing – original draft, Writing – review and editing; Jonathan Martin, Data curation, Software, Formal analysis, Validation, Investigation, Visualization, Methodology, Writing – original draft, Writing – review and editing; Aparna Uppuluri, Data curation, Validation, Investigation, Writing – original draft, Writing – review and editing; Luke R Joyce, Validation, Investigation, Writing – review and editing; Yahan Wei, Investigation; Ziqiang Guan, Resources, Supervision, Funding acquisition, Validation, Investigation, Visualization, Methodology, Writing – original draft, Writing – review and editing; Faruck Morcos, Conceptualization, Resources, Formal analysis, Supervision, Funding acquisition, Investigation, Writing – original draft, Project administration, Writing – review and editing; Kelli L Palmer, Conceptualization, Resources, Supervision, Funding acquisition, Investigation, Methodology, Writing – original draft, Project administration, Writing – review and editing

## Author ORCIDs
Priya M Christensen ⓘ https://orcid.org/0000-0003-4790-9839
Jonathan Martin ⓘ https://orcid.org/0000-0003-0946-3864
Aparna Uppuluri ⓘ https://orcid.org/0009-0005-6334-8375
Luke R Joyce ⓘ https://orcid.org/0000-0002-9346-671X
Ziqiang Guan ⓘ https://orcid.org/0000-0002-8082-3423
Faruck Morcos ⓘ https://orcid.org/0000-0001-6208-1561
Kelli L Palmer ⓘ https://orcid.org/0000-0002-7343-9271

Reviewer #1 (Public review): https://doi.org/10.7554/eLife.94929.3.sa1
Reviewer #3 (Public review): https://doi.org/10.7554/eLife.94929.3.sa2
Author response https://doi.org/10.7554/eLife.94929.3.sa3

# Additional files

## Supplementary files
• Supplementary file 1. Sequence locus IDs for all sequences listed in *Table 2*. Bold denotes confirmed *mprF* allele.

• Supplementary file 2. Fisher's exact test for determining whether positive values of hidden unit 2 are predictive of $Glc_N$-DAG specificity. p=0.028.

• Supplementary file 3. *E. coli* and plasmids used.

• Supplementary file 4. Primers used in this study, with columns indicating regions of sequence complementarity to pABG5.

• Supplementary file 5. *mprF* sequences synthesized by GeneWiz, with columns indicating regions of sequence complementarity to pABG5.

• MDAR checklist

## Data availability
All sequence data used to build the restricted Boltzmann machine model and all information used to create Figure 8 are available online through the Open Science Framework.

The following dataset was generated:

| Author(s) | Year | Dataset title | Dataset URL | Database and Identifier |
| --- | --- | --- | --- | --- |
| Christensen PM, Martin J, Uppuluri A, Joyce LR, Wei Y, Guan Z, Morcos F, Palmer KL | 2024 | Lipid discovery enabled by machine learning | https://doi.org/10.17605/OSF.IO/KM4B7 | Open Science Framework, 10.17605/OSF.IO/KM4B7 |

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
