## [Editor Report · eLife Assessment]

This study reports **important** findings on identifying sequence motifs that predict substrate specificity in a class of lipid synthesis enzymes. It sheds light on a mechanism used by bacteria to modify the lipids in their membrane to develop antibiotic resistance. The evidence is **compelling**, with a careful application of machine learning methods, validated by mass spectrometry-based lipid analysis experiments. This interdisciplinary study will be of interest to computational biologists and to the community working on lipids and on enzymes involved in lipid synthesis or modification.

---

## [Referee Report · Reviewer #1 (Public review)]

The manuscript by Christensen, et al. presents an application of restricted Boltzmann machines to analyze the MprF family of enzymes, which catalyze the addition of amino acids to lipid substrates in bacteria. Overall the manuscript is an interesting and very compelling combination of advanced statistical analysis of sequences and experimental determination of MprF function. One notable outcome is (as stated in the title) the identification of a novel substrate/product. I expect that other researchers interested in using advanced methods to connect sequence to lipid synthesis functions will find the work of significant value and that others interested in microbial resistance will find inspiration in the results. This is an excellent contribution that will be of great value to the field, and which is improved following revisions.

---

## [Referee Report · Reviewer #3 (Public review)]

Summary:

After the previous identification that the Streptococcus agalactiae MprF enzyme can synthesize also lysyl-glucosyl-diacylglycerol (Lys-Glc-DAG), besides the already known lysyl-phosphatidylglycerol (Lys-PG), the authors aim for the current manuscript was to investigate the molecular determinants of MprF lipid substrate specificity, for which MprF from a variety of bacterial species were used. This then led to the coincidental discovery of a novel lipid species.

The manuscript is well constructed and easy to follow, especially taking into account the multidisciplinary aspect of it (computational machine learning combined with lipid biology). The Restricted Boltzmann machines (RBM) approach enables the successful, although not perfect, classification and categorization of MprF activity. The computational approach is validated by lab experiments in which LC-MS analysis reveals the specific activity of the lipid synthesizing enzymes. In a few cases lipid synthesis activity is completely absent. Due to the lack of protein expression data, it is unclear if this is caused by enzyme inactivity or the overall absence of enzyme.

Overall, the authors largely achieved their goals, as the applied RBM approach led to specific sequence determinants in MprF enzymes that could categorize the specificity of these enzymes. The experimental data could largely confirm this categorization, although a stronger connection between synthesized lipids and enzyme activity would have further strengthened the observations.

The work now focuses only on MprF enzymes, but could in theory be expanded to other categories of lipid synthesizing enzymes. In other words, the RBM approach could have an impact on the lipid synthesis field, if it would be a tool that is easy applicable. Moreover, the lipids synthesized by MprF (Lys-PG, but also other cationic lipids) play an important role in the bacterial resistance against certain antibiotics.

---

## [Author Response]

The following is the authors’ response to the original reviews.

**Reviewer 1 (Public Review):**
The contribution of individual resides is shown in Figure 3c, which highlights one of the strengths of this RBM implementation - it is interpretable in a physically meaningful way. However, there are several decisions here, the justification of which is not entirely clear.i) Some of the residues in Fig 3c are stated as "relevant" for aminoacylated PG production. But is this the only such hidden unit? Or are there others that are sparse, bimodal, and involve "relevant" AA?

Thanks for bringing this important question to our attention. In fact, this was the only hidden unit involving the combination of positions 152 and 212. Although we don't have knowledge of all relevant amino acids for this catalytic process, the residues we uncover were however shown through experimental analysis to be critical for the catalytic function of two MprF variants, and thus since our protein of interest involved this function, any domain which did not contain these residues were excluded. We can't rule out that the domains we excluded from further analysis could be performing similar catalytic functions, but we found it unlikely considering the amino acids found in the negative portion of the weight were chemically unlikely to form a complex with the amino acid lysine. We have clarified in the text, that this selection is probably a subset of all important amino acids, however, this selection provided predictive power.

ii) In order to filter the sequences for the second stage, only those that produce an activation over +2.0 in this particular hidden unit were taken. How was this choice made?

The +2.0 was chosen as it ensured that the bimodal distribution was split into two distinct distributions.

iii) How many sequences are in the set before and after this filtering? On the basis of the strength of the results that follow I expect that there are good reasons for these choices, but they should be more carefully discussed.

We started with 11,507 sequences and after filtering we had 7,890 to train our model with. We think this number still maintains robust statistics. This is noted in the Dataset acquisition and pre-processing section of the Methods section.

iv) Do the authors think that this gets all of the aminoacylated PG enzymes? Or are some missed?

This is an interesting question that prompted us to do further analysis. We have added a new supplemental figure providing more details to this question. Based on the Uniprot derived annotations and the Pfam domain-based analysis of these sequences, the large majority of sequences that were excluded were proteins which included the *LPG_synthase_C* domain but **not** the transmembrane flippase domain required by the MprF class of enzymes, and were instead accompanied by different domains which seem less relevant to our enzyme of interest. It is true though, and related to question (i), that variants which might retain the functionality despite losing experimentally determined key catalytic residues could have been excluded by this method, but such sequences could still be reasonably excluded due to their dissimilarity with MprF from *Streptococcus agalactiae*.

However, some similar criticisms from the last point occur here as well, namely the selection of which weights should be used to classify the enzymes' function. Again the approach is to identify hidden unit activations that are sparse (with respect to the input sequence), have a high overall magnitude, and "involve residues which could be plausibly linked to the lipid binding specificity."(i) Two hidden units are identified as useful for classification, but how many candidates are there that pass the first two criteria? Indeed, how many hidden units are there?

We note in the Model training section of the methods that our final model used had 300 hidden units in total. As to the first part of your question, rather than systematically test the predictive power of all other hidden units to this task, we decided to use the weights that we did because of their connection to a proposed lipid binding pocket found through Autodocking experiments. While another weight might provide predictive power, it might lack this critical secondary information. Moreover, the direction of our research necessitated finding weights which first satisfied our lipid-binding pocket plausibility before using these weights to propose MprF variants to test for our novel functionality. Given the limited information we had early in the research process, to go in reverse would have provided too many options for experimental testing with reduced mechanistic justification. We included a brief explanation of our rationale in section " Restricted Boltzmann Machines can provide sensitive, rational guidance for sequence classification “ in the updated manuscript.

ii) The criterion "involve residues which could be plausibly linked to the lipid binding specificity" is again vague. Do all of the other candidate hidden units *not* involve significant contributions from substrate-binding residues? Maybe one of the other units does a better job of discriminating substrate specificity. (As indicated in Figure 8, there are examples of enzymes that confound the proposed classification.) Why combine the activations of two units for the classification, instead of 1 or 3 or...?

In fact, it is true that the other hidden units do not involve significant contribution to substrate-binding residues, and we will clarify this. The weights found through this RBM methodology are biased to be probabilistically independent, meaning that the residues and amino acids implicated by each weight are not shared among the other weights through the design of the model. We will update the Model Weight selection section to clarify that the weights we chose had more significantly weighted residues overlapping with the residues near the lipid-binding region than the other weights we checked. We combined these two because they were the only ones which had both overlap with these residues and predictive power of lipid activity with the few sequences we had detailed knowledge of at the time of decision (Figure 5b).

The Model Weight section reads as follows:

“Weights were chosen which involved sequence coordinates implicated in our function of interest. Specifically, locations identified through Autodock (Hebecker et al., 2015) where the lipid was likely to interact, and a small radius around this region to select a small set of coordinates. We chose the only weights which had both overlap with multiple residues in this chosen radius and predictive power (separation) for the three examples we had to start with.”

**Author Recommendations:**
The manuscript will likely be read by many membrane biologists/biochemists, and they might like to better understand how the RBM might be useful in their own approach. Here are some suggestions along these lines. The overall goal is to explain the RBM in *plain English* - the mathematical description in Eqs 2-4 is not easily interpretable.(1a) Explain that the RBM is a two-layer structure, in which one layer is the "visible" elements of the input sequence, and the other is called "hidden units." Connections are only made between visible and hidden units, but all such connections are made.(1b) The strengths of these connections are called "weights", and are determined in a statistical way based on a large set of input sequences. Once parametrized, the RBM is capable of capturing correlations among many positions in an input sequence - a significant advantage over the DCA approach.

We agree with this assessment, and have updated the section of the text where we introduce the RBM with a non-technical explanation of what this method is doing. It reads as:

“The design of this RBM can be seen in Figure 4, where the model architecture is represented by purple dots and green triangles. The dots are the “visible” layer, which take in input sequences and encode them into the “hidden” layer, where each triangle represents a separate hidden unit. The lines connecting the visible and hidden layers show that each hidden unit can see all the visible units (the statistics are global), but they cannot see any of the other hidden units, meaning the hidden units are mutually independent. This global model with mutually independent hidden units (see also the marginal distribution form shown in Equation 3) has the following useful properties: higherorder couplings between... “

(1c) Although strictly true that the DCA model is a Boltzmann machine, it's not a typical Boltzmann machine, because all of the units are visible. Typically a Boltzmann machine would also include hidden units, in order to increase its capacity/power.

We have clarified the relationship between DCA and Boltzmann machines, and this section now reads as:

This class of models is closely related to another model termed the Boltzmann machine. The Boltzmann machine formulation is closely related to the Potts model from physics, which was successfully applied in biology to elucidate important residues in protein structure and function (Morcos et al., 2011), and another example being the careful tuning of enzyme specificity in bacterial two-component regulatory systems (Cheng et al., 2014; Jiang et al., 2021). The Boltzmann machine-like formulation from Morcos et al. (2011), termed Direct Coupling Analysis (DCA), stores patterns...

(1d) Throughout, the authors refer to the activation of the hidden units as weights, but this is not a typical usage of this terminology. Connections between units are weights and have two subscripts. Given an input sequence, the sum over these weights for a given hidden unit is its activation (Eq. 1). I suggest aligning the description with the typical usage in order to make the presentation easier to follow. Hereafter I will refer to these hidden unit activations as simply activations.

We agree with you, the hidden units are a collection of edge weights. We have modified the terminology in the text and in our figures to consistently refer to the collections of weights as hidden units and refer to the hidden unit outputs given a sequence input as activations.

(1e) How many hidden units are there?

The final model was trained with 300 hidden units.

(2) It is redundant to say that lipids are both amphiphiles and hydrophobic...amphiphile already means hydrophobic plus hydrophilic.

This is true, we have edited the manuscript to reflect this.

(3) What does this mean, and what's the point of this remark? "They [lipids] are relatively smaller than other complex biomolecules, such as proteins, thereby allowing a larger portion of their surface to interact with other macromolecules."

We have removed this sentence.

**Reviewer 2 (Author Recommendations):**
While the idea of filtering out a part of the sequence data obtained with BLAST makes sense per se, it would be nice if the authors could comment on the nature of the sequences corresponding to the left peak in Figure 3b. It is hypothesised in conclusion that these sequences could lack any catalytic function. Could the authors experimentally check that this is the case or provide further evidence for this hypothesis?

Yes, in this revision we provide further evidence as a new supplementary figure S2. At the time we performed domain analysis of the sequences we excluded; most of these sequences lacked the flippase domain associated with MprF function, and instead were combined with different domains. On this basis we excluded them due to their lack of relevance to the MprF from *Streptococcus agalactiae* we were interested in. Although there is possibility that some relevant sequences might be excluded, our assessment is that we gained specificity by reducing the set of sequences.

A key step in the RBM-based approach is the identification of "meaningful" hidden units, i.e. whose values are related to biological function. In Methods, the authors explain how they selected these units based on the L1 norms of the weights and the region of interaction with the lipid. While these criteria are reasonable, I wonder whether they are too stringent. In particular, one could think that regions in the proteins not in direct contact with the lipid could also be important for binding. It is known for instance that the length of loops can affect flexibility and help regulate activity in some catalytic enzymes. So my question is: if one relaxes the criterion about the coordinates of large weight values, what happens? Are other potentially interesting hidden units identified?

We completely agree that other regions of the protein are likely involved in determining enzyme specificity, and that focusing on solely regions which interact with the lipid is perhaps missing important contributions to the catalytic function; we hypothesize that the flippase domain itself and its interaction with the catalytic domain are involved, especially considering the concerted mechanism by which they must operate. We are currently investigating these theories and will be the subject of future work. As an initial step, we present this current work with restricted information that led to concrete predictions. We focused on the lipid binding pocket because it was one of just a few bits of information we had from the start, but as the reviewer suggests, we plan to follow up our research to try to identify other relevant hidden units and domains.

From a purely machine-learning point of view, it would be good to see more about cross-validation of the model. More precisely, could the authors show the log-likelihood of test set data compared to the one of training sequence data?

We agree this is an important piece of information. We will update our methods section with this information. We performed a parameter sweep to search for the parameter’s we used in our final model, and in that testing with a random 80/20% training/test split we had a training log probability loss of -0.91, and a test loss of -0.98. However, for our final model we used all available data and did not perform a split; the final result did not change dramatically by including the additional data, and the weight structure and composition was consistent with the results presented in the paper.

**Reviewer 3 (Public Review):**
In many of the analyzed strains, the presence of the lipid species Lys-PG, Lys-Glc-DAG, and Lys-Glc2-DAG is correlated to the presence of the MprF enzyme(s), but one should keep in mind that a multitude of other membrane proteins are present that in theory could be involved in the synthesis as well. Therefore, there is no direct evidence that the MprF enzymes are linked to the synthesis of these lipid species. Although, it is unlikely that other enzymes are involved, this weakens the connection between the observed lipids and the type of MprF.

While there are a number of proteins found on the membrane that could play a role, we have specifically used a background strain that has a transposon in *mprF* that makes the bacteria incapable of synthesizing Lys-lipids (Figure 7B) unless complemented back with a functional MprF (Figure 7D-E). This led us to conclude that MprF is responsible for Lys-lipid synthesis.

Related to this, in a few cases MprF activity is tested, but the manuscript does not contain any information on protein expression levels. Heterologous expression of membrane proteins is in general challenging and due to various reasons, proteins end up not being expressed at all. As an example, the absence of activity for the *E. faecalis* MprF1 and E. faecium MprF2 could very well be explained by the entire absence of the protein.

The genes were expressed on the same plasmid to control for expression. While we did not run a western blot to examine expression levels the plasmid backbone was used as a control for protein expression. Previous research supports *E. faecalis* MprF1 and *E. faecium* MprF2 not synthesizing Lys-lipids and instead most likely play a different role in the cell membrane.

The title is somewhat misleading. The sequence statistics and machine learning categorized the MprFs, but the identification of a novel lipid species was a coincidence while checking/confirming the categorization.

We believe the title is appropriate given that the identification of *Enterococcus dispar* was through computational methods that led to the discovery Lys-Glc2-DAG. In other words, the categorization of potential organisms that produce lipids related to MprF has been driven by the proposition from the computational method. We agree, however, that the discovery was unexpected but would not have happened without the suggested organisms coming from the methodology presented here.

Please read the manuscript one more time to correct textual errors.The example of the role of LPS in delivering siRNA to targeted cancer cells is a bit farfetched as LPS is very different from the lipids that are being discussed here. I would rather focus on the role of Lysyl-lipids in antibiotic resistance in the introduction.

We included LPS here to explain that natural lipids/components of the bacterial cell membrane could be used for drug delivery systems. While it is true LPS is quite different from Lys-lipid compounds, our goal was to create an emphasis on how the bacterial domain is a rich untapped source of lipids that could be used in biotechnology. In this way we wanted our statement to be more broadly about bacterial lipids and the importance of their continued study for diverse applications like pharmaceuticals.

The MS identification of Lys-Glc2-DAG is convincing, especially in combination with the fragmentation data, but the ion counts suggest low abundance. The observation would be strengthened if the identification of Lysyl-Glc2-DAG with different acyl-chain configurations has been observed. This should be then mentioned or visualized in the manuscript.

We agree and have added an updated Figure 8A to demonstrate the presence of different acyl-chain configurations in *Enterococcus dispar.*

Further analysis of the Enterococcus strains shows the presence of the three lipids Lys-PG, Lys-Glc-DAG, and LysGlc2-DAG, although the Lys-Glc-DAG is only detected in trace amounts. This raises questions on the specificity of the MprF for the substrate Glc-DAG. If the ratio of Glc2-DAG compared to Glc-DAG abundance is similar to the ratio of Lys-Glc2-DAG vs. Lys-Glc-DAG abundance, this would strengthen the observation that the enzyme has equal affinity. However, if there is a rather large amount of Glc-DAG but a small amount of Lys-Glc-DAG, the production of Lys-Glc-DAG might be a side-reaction.

The reviewer brings a relevant point of discussion, however, a clear resolution might be part of future work as we do not use spike in controls when completing lipid extractions. Because of this, it it is not possible for us to compare lipid levels across different samples. We now include a note clarifying this in the discussion section.

The plotting of the MprF sequence variants using the chosen RBM weights reveals a rather complex distribution over the quadrants (Figure 8). It is rather unclear in Figure 8 why only 1 sequence is plotted for *Enterococcus faecalis* and faecium, while 2 different MprFs are present (and tested) for these two organisms. This should be clarified.

We agree this can be a source of confusion. We have further clarified this in the text that only the functional alleles were plotted in Figure 8 and that all Enterococcal alleles are plotted in Figure S3 regardless of function.